# PDGFRβ Activation Induced the Bovine Embryonic Genome Activation via Enhanced NFYA Nuclear Localization

**DOI:** 10.3390/ijms242317047

**Published:** 2023-12-01

**Authors:** Chalani Dilshani Perera, Muhammad Idrees, Abdul Majid Khan, Zaheer Haider, Safeer Ullah, Ji-Su Kang, Seo-Hyun Lee, Seon-Min Kang, Il-Keun Kong

**Affiliations:** 1Division of Applied Life Science (BK21 Four), Gyeongsang National University, Jinju 52828, Republic of Korea; chalanidilshani99@gmail.com (C.D.P.); idrees1600@gmail.com (M.I.); abdulmajid3042@gmail.com (A.M.K.); zaheerhdr@gmail.com (Z.H.); khan00safeer@gmail.com (S.U.); tltn0728@gmail.com (J.-S.K.); womxnking@gmail.com (S.-H.L.); sunmin0810@gmail.com (S.-M.K.); 2Institute of Agriculture and Life Science (IALS), Gyeongsang National University, Jinju 52828, Republic of Korea; 3The King Kong Corp. Ltd., Gyeongsang National University, Jinju 52828, Republic of Korea

**Keywords:** embryonic genome activation, NFYA, PDGFRβ, bovine blastocysts, activator

## Abstract

Embryonic genome activation (EGA) is a critical step during embryonic development. Several transcription factors have been identified that play major roles in initiating EGA; however, this gradual and complex mechanism still needs to be explored. In this study, we investigated the role of nuclear transcription factor Y subunit A (NFYA) in bovine EGA and bovine embryonic development and its relationship with the platelet-derived growth factor receptor-β (PDGFRβ) by using a potent selective activator (PDGF-BB) and inhibitor (CP-673451) of PDGF receptors. Activation and inhibition of PDGFRβ using PDGF-BB and CP-673451 revealed that NFYA expression is significantly (*p <* 0.05) affected by the PDGFRβ. In addition, PDGFRβ mRNA expression was significantly increased (*p <* 0.05) in the activator group and significantly decreased (*p <* 0.05) in the inhibitor group when compared with PDGFRα. Downregulation of NFYA following PDGFRβ inhibition was associated with the expression of critical EGA-related genes, bovine embryo development rate, and implantation potential. Moreover, ROS and mitochondrial apoptosis levels and expression of pluripotency-related markers necessary for inner cell mass development were also significantly (*p <* 0.05) affected by the downregulation of NFYA while interrupting trophoblast cell (*CDX2*) differentiation. In conclusion, the PDGFRβ-NFYA axis is critical for bovine embryonic genome activation and embryonic development.

## 1. Introduction

Early gestation is the most sensitive period of embryonic development, with approximately 60% of embryonic mortality occurring during the early developmental stages [1]. Embryonic genome activation (EGA) is an important event where the newly generated embryonic genome drives the differentiation of all cell types in adults [2]. Embryonic development begins with a single cell and is generated after the fusion of two haploid cells, a sperm cell, and an oocyte that encodes a single zygotic genome [3]. Upon fertilization, the zygotic genome is inactive transcriptionally, and the newly formed zygote can achieve a totipotent state and produce an entirely new organism. However, EGA is required for this critical step [4]. The initiation of the first transcription after fertilization in newly formed embryos is denoted as EGA, and the first transcripts can be detected in a species-specific period [5]. The embryonic genome is gradually activated by maternal-to-zygotic genome transition (MZT), which transfers the control of embryonic development from maternal deposits to zygotic factors [6]. Maternal mRNAs and proteins are depleted [7]. The MZT is important for preparing embryos for cell differentiation and further development. Therefore, MZT coordinates EGA and cell division [8]. Zygotic transcription is gradually activated after a specific number of cell cycles as a part of the MZT [9]. Transcriptional activation occurs in two waves, a minor wave and a major wave. During early cleavage stages, a minor wave occurs, and the major wave constitutes widespread transcriptional activation of zygotic genes coinciding with the lengthening of the cell cycle [2,10]. According to the animal species, the timing of these waves and number of cell division cycles varied.

Species-specific activation of the embryonic transcriptional program relies on key transcription factors [11]. Several transcriptional factors that initiate EGA and transcription of the embryonic genome have been identified. Zelda (ZLD), POU domain, class 5, transcription factor 3 (POU5F3), SRY-box transcription factor 19b (SOX19B), Nanog homeobox (NANOG), and double homeobox (DUX) are the master regulators of EGA in mice, zebrafish, flies, and humans [10]. ZLD was the first key activator of EGA identified in Drosophila, and no ZLD homolog has been discovered outside of this insect group [12]. The POU5F3, SOX19B, and NANOG transcription factors are maternally deposited as mRNA [13]. POU5F3, SOX19B, and NANOG, which regulate EGA, are homologs of three key transcription factors (organic cation/carnitine transporter4 (OCT4), SOX2, and NANOG) that maintain stem cell pluripotency [10]. In addition to these transcription factors, nuclear transcription factor Y subunit A (NFYA) was recently shown to play a role in EGA in mammals [14]. NFYA is the regulatory subunit of the NFY complex and has sequence-specific DNA-binding ability. Binding to DNA forms a histone-like structure and promotes chromatin accessibility [15]. NFYA is a CCAAT-binding transcription factor. In addition, NFYA binds to target genes’ promoter or enhancer regions [16]. Furthermore, NFYA is essential for activating a certain subset of genes during the major wave of EGA and is required to develop the blastocyst stage [17]. According to Lu et al. (2016), the knockdown of NFYA in mice downregulated the two-cell embryo-activated genes by ~15%. It revealed that the NFYA transcription factor is important for EGA in mammals [15]. The loss of function of this transcription factor directly affects transcriptional activation and embryonic development [18]. However, further investigations of the NFYA transcription factor are required.

Some cell surface receptors mediate the activity of NFYA transcription factors [19]. PDGFRβ is a cell surface receptor that belongs to the platelet-derived growth factor (PDGF) family [20]. PDGF signaling is crucial in embryogenesis [21]. A prior study showed that PDGFRβ is the upstream regulator of NFYA and, via mitogen-activated protein kinases (MAPK), can localize NFYA inside the nucleus [22]. Moreover, the three subunits of NFY (NFYA, NFYB, and NFYC) control PDGFR expression at various cell cycle phases [21]. However, the relationship between PDGFRβ and the NFYA transcription factor has not been studied in early bovine embryonic development. The role of NFYA is poorly understood in bovine EGA.

This study explored the roles of NFYA and PDGFRβ in bovine embryonic genome activation. We activated and inhibited PDGFRβ via a potent, selective activator and inhibitor to understand its link with NFYA and bovine EGA. Furthermore, we analyzed the embryonic development rate, implantation potential, and apoptotic signals in PDGFRβ activator and inhibitor-treated bovine embryos. We also analyzed the popular markers of EGA and stem cells to identify the effects of both the PDGFRβ and NFYA on bovine embryonic development.

## 2. Results

### 2.1. Dynamic Changes in PDGFRβ and NFYA Expression during Bovine Embryonic Development

First, we analyzed NFYA, PDGFRβ, and PDGFRα localization at all stages of embryonic development via immunofluorescence. We found that PDGFRβ and PDGFRα were cytoplasmically localized in oocytes surrounding cumulus cells and from the zygotic stage to the blastocyst stage. In contrast, NFYA was localized in the nucleus from the single-cell zygote to the day-8 blastocyst (Figure 1A). The nuclear localization of NFYA from the pronuclear stage to the blastocyst stage suggests that it may be involved in the embryonic genome activation of bovine embryos. Next, we used RT-qPCR to analyze the mRNA expression of *NFYA*, *PDGFRβ*, and *PDGFRα* from the MII to day-8 bovine blastocysts. According to the findings, all the genes are expressed from the MII stage to the blastocyst stage (Figure 1B).

### 2.2. Effect of PDGFRβ on NFYA and Embryonic Development

We hypothesized that PDGFRβ is the upstream regulator of NFYA. To examine the link between PDGFRβ and NFYA, we activated and inhibited PDGFRβ via a potent, selective activator (PDGF-BB) and inhibitor (CP-673451) of PDGF receptors. We found that after PDGF-BB and CP-673451 treatment, PDGFRβ mRNA expression was significant (*p <* 0.05) among control, activator, and inhibitor groups when compared with PDGFRα (Figure 2B). Therefore, studies were continued only for the PDGFRβ receptor and NFYA. We found that the activation and inhibition of PDGFRβ significantly (*p <* 0.05) affected NFYA expression and nuclear localization in 8- and 16-cell embryos and day-8 bovine blastocysts (Figure 2A,B). PDGF-BB-treated bovine embryos have highly localized NFYA in the nucleus compared with the control group, and CP-673451-treated embryos have significantly (*p <* 0.05) lower NFYA protein levels in the nuclei. Furthermore, other components of the NFY transcription factor, such as *NFYB* and *NFYC*, were also examined in the activator and inhibitor groups. These also showed differential expressions in all groups but were not as significant as NFYA (Figure 2C). Next, we examined the effects of activation and inhibition of PDGFRβ on bovine embryonic development (Table 1) (Figure 2D). We found that compared with the control group (30.0%), activator-treated embryos had a significantly (*p <* 0.05) higher developmental rate (38.7%), while inhibitor-treated embryos had a significantly (*p <* 0.05) lower developmental rate (16.8%).

### 2.3. PDGFRβ and NFYA Are Linked to Embryonic Genome Activation

The above results suggest that PDGFRβ regulates NFYA. Therefore, we examined the role of NFYA in bovine embryonic genome activation. Genes related to embryonic genome activation, such as *DUXA*, *GSC*, *ARGFX*, *SP1*, and *DPRX*, were examined in the PDGFRβ activator and inhibitor groups. Critical genes involved in EGA were significantly (*p <* 0.05) and differentially expressed in the activator and inhibitor groups (Figure 3A). NFYA is the direct transcription factor of serum response factor (*SRF)* and Zinc fingers and homeoboxes 1 (*ZXH1*); therefore, we also examined the mRNA expression of these genes. Both genes were significantly (*p <* 0.05) and differentially expressed with PDGFRβ-linked NFYA activation and inhibition (Figure 3B). Next, to examine the role of NFYA in early embryonic cellular differentiation, we examined trophoblast (CDX2) and inner cell mass (OCT4/NANOG/SOX2/KLF4/SALL4) markers. We found that PDGF-BB-treated embryos exhibited significantly (*p <* 0.05) high mRNA expression levels of *CDX2* (Figure 3C). Moreover, the activator group showed significantly (*p <* 0.05) higher mRNA expression levels of *OCT4*, *NANOG*, *SOX2*, *KLF4,* and *SALL4* than the control and inhibitor groups (Figure 3D).

### 2.4. MAPK1/AKT and JAK/STAT Pathway Roles in NFYA Nuclear Localization via PDGFRβ Activation

To analyze the PDGFRβ-linked NFYA nuclear localization, we examined NFYA protein expression via immunofluorescence. The results showed that PDGFRβ activation significantly (*p <* 0.05) enhanced NFYA nuclear localization in 8- and 16-cell stages and day-8 bovine blastocysts, compared to the control and PDGFRβ inhibition group (Figure 2A). The above result suggests that downstream signaling of PDGFRβ activates the NFYA. We first examined the MAPK1/AKT pathway. Gene expression analysis showed that *MAPK1* and *AKT* were significantly (*p <* 0.05) upregulated in the PDGFRβ activator group (Figure 4A). Furthermore, we examined another most prominent JAK/STAT downstream pathway of the PDGFRβ receptor. The mRNA expression of *JAK2* significantly (*p <* 0.05) showed enhancement while upregulating the *STAT3* mRNA expression in the PDGFRβ activator group compared to the control and inhibitor groups (Figure 4B). Furthermore, we also examined transcription factors that have interacted and linked with NFYA, such as CIITA and CREB. The mRNA expressions of the *CIITA* and *CREB* were significantly (*p <* 0.05) higher in the PDGFRβ activator group (Figure 4C).

### 2.5. PDGFRβ and NFYA Effects on Embryonic Mitochondria and Implantation Potential

The mitochondrial genome is activated after minor embryonic genome activation and plays an important role in the full flag activation of EGA. We examined mitochondrial biogenesis in PDGFRβ activator and inhibitor-treated bovine embryos. PDGFRβ-linked differential NFYA expression significantly (*p <* 0.05) regulates mitochondrial membrane potential (Figure 5A). In comparison to the control and inhibitor groups, the activator group had a significantly (*p <* 0.05) higher number of mitochondrial aggregates. Furthermore, ROS and mitochondrial apoptosis levels were also significantly (*p <* 0.05) decreased in the PDGFRβ activator group compared with the control and inhibitor groups (Figure 5B,C). Next, we examined the importance of NFYA in embryo implantation using an invasion assay. A comparison of the activator group with the control and inhibitor groups showed that the invasion area and proliferation of implanted blastocysts were significantly (*p <* 0.05) higher in the activator group (Figure 5D). These results showed that PDGFRβ activation and inhibition significantly (*p <* 0.05) affected apoptosis and embryonic cell numbers.

## 3. Discussion

EGA is a critical molecular event that occurs during the early stages of embryonic development and is necessary for the development of an organism. However, the key regulators and molecular processes that induce EGA are poorly understood in most animals [23]. Several transcription factors play important roles in initiating EGA and determining the fate of blastomeres during the early stages of embryonic development [24]. A previous study showed that NFYA, an NFY complex subunit, contributes to the mouse EGA [15]. According to a previous study, NFYA is the downstream regulator of the PDGFRβ receptor, and its activation translocates NFYA to the nucleus for transcription-related activities [22]. No previous studies have explored the expression and role of NFYA in bovine embryonic genome activation through inhibition and activation of the PDGFRβ receptor. In this study, we demonstrated that activation and inhibition of the PDGFRβ receptor significantly (*p <* 0.05) affected the expression of NFYA in bovine embryos and revealed that PDGFRβ controls NFYA. Our study suggests that reducing NFYA expression through inhibition of PDGFRβ significantly (*p <* 0.05) affects the initiation of bovine embryonic genome activation and bovine blastocyst development.

According to a previous study, NFYA is expressed at all stages of mouse preimplantation development, indicating that NFYA is maternally stored [25]. This study also found that NFYA was expressed in COCs up to day 8 in bovine blastocysts, indicating maternal expression. However, *NFYA* expression levels in bovine and mouse preimplantation embryos differ [25]. As previously mentioned, PDGFRβ is the upstream regulator of NFYA, and we found that PDGFRβ was expressed in all stages of early bovine embryonic development [22]. Kim and Fischer stated that PDGFRβ plays an important role in the embryonic development of bovines but did not reveal the mechanism of PDGFRβ in embryonic development [26]. In addition, we found that NFYA was localized in the nucleus as a crucial regulator in EGA, and PDGFRβ was localized in the cytoplasm [27,28,29]. To prove that PDGFRβ and NFYA have the same link as previously mentioned, we activated and inhibited PDGFRβ and found that NFYA was highly affected by PDGFRβ [22].

According to a previous study, RNA-seq data of NFYA knockdown two-cell embryos demonstrated that NFYA is necessary for activating 15% of two-cell-activated genes, indicating that NFYA is responsible for EGA in mice [15]. Furthermore, NFYA contributes to activating a subset of genes in the major wave of EGA and is important for progression to the blastocyst stage [24]. Moreover, a previous study stated that NFYA regulates the expression of transcriptional regulatory genes during the embryonic genome activation in zebrafish embryos [30]. Previous studies have shown that SP1, GSC, DUXA, ARGFX, and DPRX were involved in mammalian EGA [31,32,33,34]. ARGFX and DPRX are newly identified transcription factors involved in EGA [34]. SP1 and DUXA directly interact with the NFYA [35,36,37]. Additionally, previous studies poorly understood the direct or indirect relationship between PDGFRβ activation and these genes independently from NFYA. Therefore, our study provides an approach to elucidate the upregulation of NFYA via PDGFRβ activation associated with the expression of these EGA genes. Furthermore, we found that NFYA depletion reduced the expression of *SRF* and *ZXH1*. Previous studies have shown that SRF and ZXH1 are directly associated with NFYA [38]. Furthermore, the knockdown of NFYA in embryonic stem cells (ESC) results in severe proliferation abnormalities and a significant decline in the expression of *OCT4*, *NANOG*, and other pluripotency genes [39]. Based on the results of previous studies, we identified genes related to pluripotency, including *OCT4*, *NANOG*, *SOX2*, *KLF4,* and *SALL4*, that were downregulated by the differential expression of NFYA [40,41]. In addition to controlling cell pluripotency, these genes are also involved in mammalian EGA [42]. Therefore, the downregulation of these genes inversely affects bovine EGA. In bovine blastocysts, the homeobox gene *CDX2* regulates several trophoblast genes [43]. *CDX2* downregulation affects the timing of blastocoel formation and cell proliferation during the morula to blastocyst stage. [44]. We found that reducing NFYA through PDGFRβ inhibition also affected *CDX2* expression in trophoblast cells. Interestingly, NFYA contributes to the regulation of pluripotency and trophectoderm-related markers. PDGFRβ activation and inhibition significantly affected bovine embryonic development, demonstrating the role of NFYA in EGA and its connection to PDGFRβ. Through the MAPK1/AKT pathway, we confirmed the relationship between NFYA and PDGFRβ [22].

Mitochondria play a crucial role in early embryonic development in mammals [45]. Several studies have claimed that the mitochondrial genome is activated even before the major wave of EGA and that several maternal factors that play a role in EGA also play an effective role in mitochondrial genome activation [18,46]. Furthermore, mitochondrial enzymes play important roles in EGA and embryonic development [47]. A mitochondrial malfunction activates apoptotic cascades, resulting in aberrant embryonic development and other abnormalities [48]. Our study found that depletion of NFYA expression through PDGFRβ inhibition reduces the mitochondrial membrane potential and enhances the monomeric form of mitochondria. A previous study found that sequence-specific subunit NFYA knockdown triggers defects in S-phase progression, leading to apoptotic cell death [49]. The inhibition of PDGFRβ by CP-673451 significantly increased apoptosis, which is associated with ROS accumulation [50]. Moreover, we found that reducing NFYA through PDGFRβ inhibition was associated with implantation potential and mitochondrial activity in bovine embryos [51].

## 4. Materials and Methods

This study was performed in accordance with the instructions of the Institutional Animal Care Committee of Gyeongsang National University (GNU-230425-A0088). Unless otherwise stated, all reagents used in this study were purchased from Sigma-Aldrich (St. Louis, MO, USA).

### 4.1. Experimental Design

#### 4.1.1. Experiment 1 (PDGFRβ Activation)

The PDGFRβ activator PDGF-BB (cat. # Z03707-10) was added to the in vitro culture (IVC) medium. The cumulus–oocyte complexes (COCs) were separated into five experimental groups to optimize the PDGF-BB concentration. For the embryo culture, the presumed zygotes were pipetted and removed the cumulus cells. Then, the denuded zygotes were treated with 0 (control group), 20, 30, 50, and 100 ng/mL PDGFR-BB in an IVC medium for three days considering In Vitro Fertilization (IVF) as day 0. After three days, IVC media was replaced without adding an activator or inhibitor. PDGF-BB (50 ng/mL) was selected as the optimal concentration based on the blastocyst development rate at day 8. Five biological replicates were used to determine the optimal (effective) concentrations. PDGFRβ and NFYA protein and gene expression levels were analyzed in 8-cell-stage embryos, 16-cell-stage embryos, and day-8 blastocysts through immunofluorescence and real-time quantitative polymerase chain reaction (qRT-PCR). Apoptotic signals and reactive oxygen species (ROS) levels were analyzed using the terminal deoxynucleotidyl transferase (TdT) dUTP Nick-End Labeling (TUNEL) and ROS assays. The mRNA expression levels of genes related to EGA, stem-cell markers, and trophoblast cell development-associated genes were analyzed by qRT-PCR using 8-cell-stage embryos, 16-cell-stage embryos, and day-8 blastocysts. Embryo quality was evaluated using an in vitro invasion assay and 5,5,6,6′-tetrachloro-1,1′,3,3′ Tetraethylbenzimidazolylcarbocyanine iodide (JC-1) staining.

#### 4.1.2. Experiment 2 (PDGFRβ Inhibition)

Experiment 2 was conducted similarly to the experiment 1. The specific inhibitor CP-673451 (cat. # S 1536) was added at different concentrations (0 (control group), 100, 200, and 500 nM) to the IVC medium. Following comparison with the control group, 100 nM was determined to be the lowest effective concentration based on the blastocyst developmental rate. Five biological replicates were used to determine the optimal (effective) concentrations. After determining the optimal concentration of the inhibitor, all the experimental procedures were identical to those performed in the PDGFRβ activation.

### 4.2. Oocyte Collection and In Vitro Maturation (IVM)

Oocyte collection and in vitro maturation processes were conducted according to the available literature [52]. Bovine ovaries were retrieved from a nearby slaughterhouse and delivered to the laboratory in a container with sterile saline at 37 °C in less than 2 h. The ovaries were washed with warm Dulbecco’s phosphate-buffered saline (DPBS). Then, Tyrode’s lactate HEPES (TL-HEPES) medium (sodium bicarbonate 2 mM (s-5761), sodium chloride 114 mM (s-5886), sodium lactate 10 mM, sodium phosphate monobasic 0.34 mM (s-5011), potassium chloride 3.2 mM (p-5405), phenol red 1 µL/mL, calcium chloride 2 mM (c-7902), magnesium chloride 0.5 mM (M-2393), streptomycin 0.1 mg/mL solution, and penicillin 0.1 mg/mL) were used to collect COCs from 3–8 mm follicles. After aspiration, COCs with homogeneous cytoplasm and cumulus cells with more than three layers were collected and washed four times with TL-HEPES medium. Thereafter, approximately 50 COCs were cultured in 4-well plates (Nunc, Roskilde, Denmark) containing 600 µL IVM medium (Medium 199, Thermo Fisher Scientific, Waltham, MA, USA; cat. # 11150059) supplemented with 10% (*v*/*v*) fetal bovine serum (FBS; Gibco BRL, Life Technologies, Grand Island, NY, USA; cat. # 16000-044), 0.6 mM cysteine (cat. # c7352), 1 µg/mL estradiol-17β (cat. E-2758), 0.2 mM sodium pyruvate (cat. # p-5280), 10 ng/mL epidermal growth factor (EGF) (cat. # E-4127), and 10 µg/mL follicle-stimulating hormone (FSH) (Prospec, East Brunswick, NJ, USA, cat. # HOR-285) under 5% CO_2_ in a humidified environment at 38.5 °C for 22–24 h.

### 4.3. In Vitro Fertilization (IVF) and Culture (IVC) of Embryos

As previously mentioned, frozen–thawed bovine sperm fertilized in vitro-matured oocytes in IVM media [53]. Briefly, the same batch of frozen semen straws was diluted in DPBS after thawing at 37.5 °C for 1 min. The sperm suspension was centrifuged by 750× *g* at 27 °C for 5 min. The pellet was resuspended in 500 μL heparin (1.75 µg/mL) in IVF medium (Penicillin (100 IU/mL), sodium pyruvate (22 mg/mL), streptomycin (0.1 mg/mL), and BSA (6 mg/mL), added to the (TL-HEPES solution). To aid capacitation, sperm were incubated at 38.5 °C in a humid environment with 5% CO_2_ air for 15 min. Subsequently, sperm treated with heparin were diluted in the IVF medium. (1 × 10^6^ sperm/mL as the final density). Then, the matured COCs were cultured in 500 μL IVF medium at 38.5 °C in a humid environment with 5% CO_2_ for 20 h. By repeatedly pipetting, the cumulus cells were removed from the oocytes after co-culturing with spermatozoa for 20 h. Following that, presumptive zygotes were denuded and washed four times and cultured in 4-well dishes consisting of 600 μL synthetic oviductal fluid (SOF) medium composed of 5 ng/mL sodium selenite (cat. # 11074547001), 5 µg/mL transferrin, 5 µg/mL insulin, 4 mg/mL fatty acid-free BSA, and 100 ng/mL epidermal growth factor (EGF) supplemented.

### 4.4. Immunofluorescence Analysis and Antibodies

Confocal microscopy was performed to determine protein expression and localization in the samples [52]. COCs, mature oocytes (MII)/zygotes/two-cell/four-cell/eight-cell/16-cell stage embryos, and day-8 blastocysts were collected from the control activator and inhibitor groups. The collected embryos were fixed with 4% (*w*/*v*) paraformaldehyde in 1.0 M PBS. Fixed embryos were washed thrice with polyvinyl alcohol (0.3%) in 1× PBS (PVA PBS) for 15 min. Proteinase K was added to the samples and incubated for 5 min to enhance permeability. The samples were washed three times with PVA in PBS for 15 min and incubated with the blocking solution (BSA 5% in PBS/PVP) at 27 °C for 90 min. Then, the samples were incubated overnight with PDGFRβ (Santa Cruz Biotechnology Inc., Dallas, TX, USA; cat. # sc-373805), NFYA (My Biosource.com, San Diego, CA, USA, cat. MBS822279) primary antibodies at 4 °C. The samples were thoroughly washed the following day with PVA-PBS for 15 min. Thereafter, the samples were incubated with TRITC and FITC secondary antibodies (Santa Cruz Biotechnology, Dallas, TX, USA) at 27 °C for 90 min and washed thrice with PVA PBS for 15 min to remove residual stains. To stain nuclei, 4, 6-diamidino-2-phenylindole (DAPI, 10 g/mL) was added to the samples at 27 °C for 5 min. After washing, all samples were fixed on slides. A laser scanning confocal microscope (Fluoview FV 1000; Olympus, Tokyo, Japan) was used for confocal imaging. Signal intensities were measured using ImageJ 154 software (National Institutes of Health, Bethesda, MD, USA; https://imagej.nih.gov/ij, accessed on 1 September 2022).

### 4.5. Detection of Embryonic Reactive Oxygen Species (ROS)

As previously mentioned, blastocyst ROS levels were measured using a ROS assay [52]. Briefly, PBS containing 10 nM 2’,7′-dichlorofluorescein diacetate (H_2_DCFDA) (cat. # D6883) was used to incubate live blastocysts at 38.5 °C and humidified environment with 5% CO_2_ for 30 min. Following incubation, the blastocysts were thoroughly washed three times with PVA and PBS for 15 min. The samples were then viewed using an epifluorescence microscope (Olympus IX71, Tokyo, Japan) with 490 nm excitation and 525 nm emission filters.

### 4.6. TUNEL Assay

As prior studies mentioned, a TUNEL assay was performed to evaluate the apoptotic index of day-8 bovine blastocysts using a cell death detection kit (in situ) (TMR red, Roche Diagnostics, Mannheim, Germany, cat. #12156792910) according to manufacturer’s instructions [1]. Embryos were fixed in 4% (*w*/*v*) paraformaldehyde and washed thrice with PVA PBS for 15 min. After washing, day-8 blastocysts were incubated for 1 h with the TUNEL assay kit reagents at 38.5 °C and 5% CO_2_ under dark conditions. Thereafter, the blastocysts were washed with PVA and PBS for 15 min. The cells were then stained with DAPI (10 g/mL) at 27 °C for 5 min to detect the nuclei. The DAPI-stained blastocysts were washed again with PVA in PBS for 15 min and fixed on slides. Subsequently, a mercury lamp-equipped epifluorescence microscope (Olympus IX71) was used to count DAPI-stained cells and determine the number of cells per blastocyst. Healthy cells are shown in blue, whereas TUNEL-positive apoptotic cells are shown in bright red.

### 4.7. JC-1 Staining

The mitochondrial membrane potential (Ψ) was determined by JC-1 staining of day-8 blastocysts. Live day-8 blastocysts were washed thrice with PVA in PBS for 15 min [52]. The blastocysts were then stained with JC-1 (2.0 μg/mL) (cat. # T3168) diluted in D-PBS with 5% CO_2_ at 38.5 °C for 30 min in the dark. Following incubation, the blastocysts were washed thrice with a PVA-PBS solution for 15 min and then stained with DAPI for 5 min. The washed blastocysts were fixed on slides, and images were viewed under a laser scanning confocal microscope (FV1000, Olympus, Tokyo, Japan).

### 4.8. Invasion Assay

An invasion assay measured the invasion area and proliferation of day-8 blastocysts. First, cell culture inserts (6.4 mm; Corning Inc., Corning, NY, USA) were placed in 24-well plates (Corning Inc., Corning, NY, USA) [52]. Thereafter, Matrigel (20 mg per filter; Discovery Labware Inc., Bedford, MA, USA) was applied to the upper surface of the chamber and incubated at 38.5 °C to allow drying for 30 min. The blastocysts were transferred to a Matrigel-coated filter (each culture insert contained three blastocysts in the same media (SOF medium) used for embryo culture) and cultured at 38.5 °C in a humid environment with 5% CO_2_ for 72 h. Every 48 h, the medium was replaced with a new one. The cells were stained with DAPI for 5 min. The proliferation and invasion areas of trophoblasts were assessed after ten days of culture using an Olympus IX71 microscope. ImageJ software 154 was used to analyze the data.

### 4.9. mRNA Extraction of and Complementary DNA (cDNA) Synthesis Real-Time Quantitative PCR (qRT-PCR) Analysis

To analyze gene expression, RNA was isolated from fifteen COCs, twenty MII stage oocytes/zygotes/2-cell/4-cell stage embryos, fifteen 8-cell, 16-cell embryos, and six day-8 blastocysts per replicate using an Applied Biosystems^TM^ PicoPure^TM^ RNA isolation kit according to the manufacturer’s instructions (Thermo Fisher Scientific, Ottawa, ON, Canada) [52]. The concentration of RNA and quality were measured using a Nanodrop spectrophotometer (2000c NANO DROP, Thermo Fisher Scientific) at 260 nm. Thereafter, cDNA was synthesized by reverse transcription using an iScript cDNA synthesis kit (Bio-Rad Laboratories, Hercules, CA, USA; cat. #1708891) in a total volume of 20 µL according to the manufacturer’s guidelines.

As explained previously, real-time qPCR was conducted to quantify gene expression levels using a CFX98 instrument (Bio-Rad Laboratories, Hercules, CA, USA) [52]. First, the qRT-PCR primers for the genes of interest and reference/housekeeping genes were obtained from Integrated DNA Technologies (Coralville, IA, USA). The reaction was performed in a final volume of 10 µL consisting of 5 µL from the iQ SYBR Green Supermix kit (Bio-Rad Laboratories; cat. # 170-8882), 3 µL of diluted cDNA, and 1 µL of forward and reverse primers, which were bovine-specific. Each reaction was performed in triplicate, and non-template (with added nuclease-free water) samples were included as negative controls. Glyceraldehyde-3-phosphate dehydrogenase (*GAPDH*) was used as the reference gene and internal control to normalize the expression data. The following PCR cycle conditions were used: initial denaturation (95 °C for 3 min), followed by 44 cycles of 95 °C for 15 s, 57 °C for 20 s, 72 °C for 30 s, and a final extension at 72 °C for 5 min and primer extension (at 72 °C for 5 min). The relative gene expression levels were measured in relation to a gene of interest and a reference gene based on the ‘delta delta’ (^ΔΔ^Ct) values. Primer information is listed in Table 2.

### 4.10. Statistical Analysis

GraphPad Prism version 8.0 (GraphPad Software, Franklin Street, Boston, MA, USA; www.graphpad.com, was used to analyze the data. The collected data were evaluated using one-way and two-way ANOVA. The density values are expressed as the mean ± standard error. For all comparisons, *p <* 0.05 was considered statistically significant, and the mean was used to express the numerical data ± SD.

## 5. Conclusions

In conclusion, we found that the activation and inhibition of PDGFRβ significantly and differentially regulated the NFYA transcription factor in bovine embryos. Downregulation of NFYA through PDGFRβ inhibition reduces embryonic genome activation and the expression of genes related to this process. Activation of PDGFRβ significantly enhanced the development rate of bovine embryos and their implantation potential. Moreover, the expression of pluripotent stem cell markers (*OCT4*, *NANOG*, *SOX2*, *KLF4,* and *SALL4*) necessary for inner cell mass development and trophoblast cell development was also significantly affected by NFYA. Our study collectively revealed that the PDGFRβ-NFYA axis is critical for bovine embryonic genome activation and healthy embryo development. This approach could be beneficial for establishing more effective protocols for in vitro embryo production in the future.

## Figures and Tables

**Figure 1 ijms-24-17047-f001:**
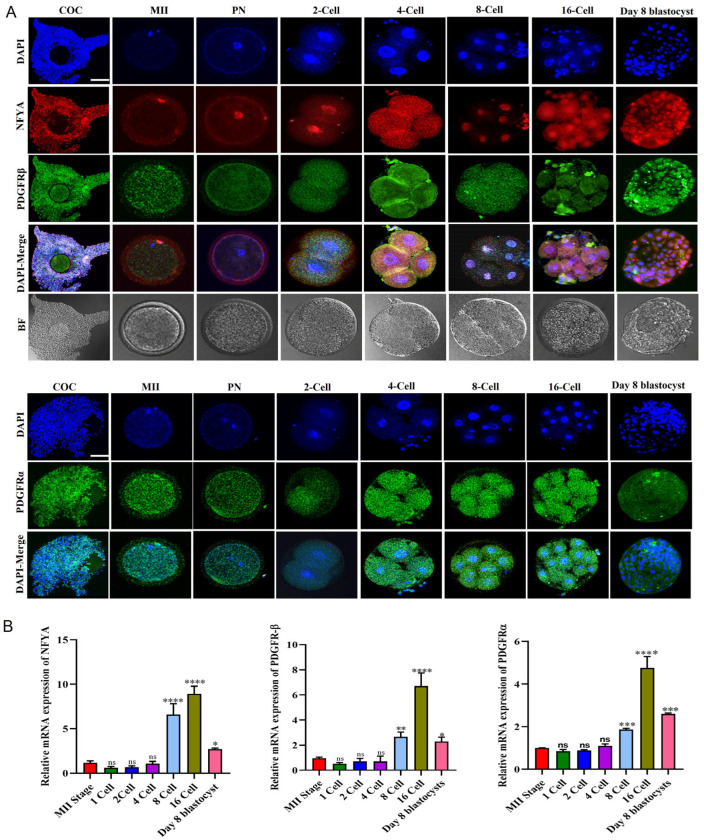
Protein expression and mRNA expression of NFYA, PDGFRβ, and PDGFRα during bovine embryonic development. (**A**) Immunofluorescent expression of NFYA, PDGFRβ, and PDGFRα from COCs to day-8 blastocyst stage. (**B**) Relative mRNA expression of *NFYA*, *PDGFRβ*, and *PDGFRα* from MII to day-8 blastocyst stage. Data are presented as the mean ± SEM. For immunofluorescent staining, COCs/MII stage oocytes/zygotes/two cell/four cell/8-cell/16-cell-stage embryos and day 8 blastocysts, 20 per group, were used. For mRNA expression analysis, MII stage oocytes/zygotes/two-cell/four-cell-stage embryos, 20 per group; 8-cell and 16-cell embryos, 15 per group, and 6 blastocysts per group were used in triplicates. The experiments were repeated three times. Scale bar: 20 µm. ns = not significant. * *p <* 0.05, ** *p <* 0.01, *** *p <* 0.001, and **** *p <* 0.0001 denote significant differences; SEM—standard error of mean. BF = Bright Field.

**Figure 2 ijms-24-17047-f002:**
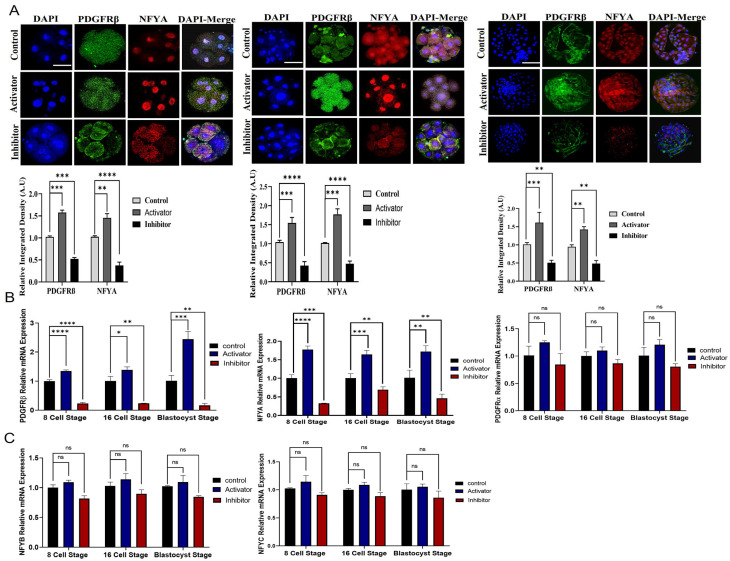
PDGFRβ activation and inhibition effects on NFYA. (**A**) Immunofluorescent expression of PDGFRβ and NFYA in 8-cell embryos, 16-cell embryos, and day-8 bovine blastocysts control, activator, and inhibitor groups. (**B**) Relative *PDGFRβ*, *NFYA,* and PDGFRα gene expression through RT-qPCR in 8-cell embryos, 16-cell embryos, and day-8 bovine blastocysts of control, activator, and inhibitor groups. (**C**) Relative mRNA expression of *NFYB* and *NFYC* in 8-cell embryos, 16-cell embryos, and day-8 blastocyst control, activator, and inhibitor groups. (**D**) Dose-dependent response of PDGF-BB and CP-673451 on embryo cleavage percentage and blastocyst developmental percentage. Data are presented as the mean ± SEM. For mRNA analysis, 8-cell and 16-cell embryos, 15 per group, and 6 blastocysts per group were used in triplicates. For immunofluorescent staining, n = 20 8-cell and 16-cell embryos and n = 15 blastocysts per group were used. The experiments were repeated three times. Scale bar: 20 µm. ns, not significant, * *p <* 0.05, ** *p <* 0.01, *** *p <* 0.001, and **** *p <* 0.0001 denote significant differences. SEM—standard error of mean.

**Figure 3 ijms-24-17047-f003:**
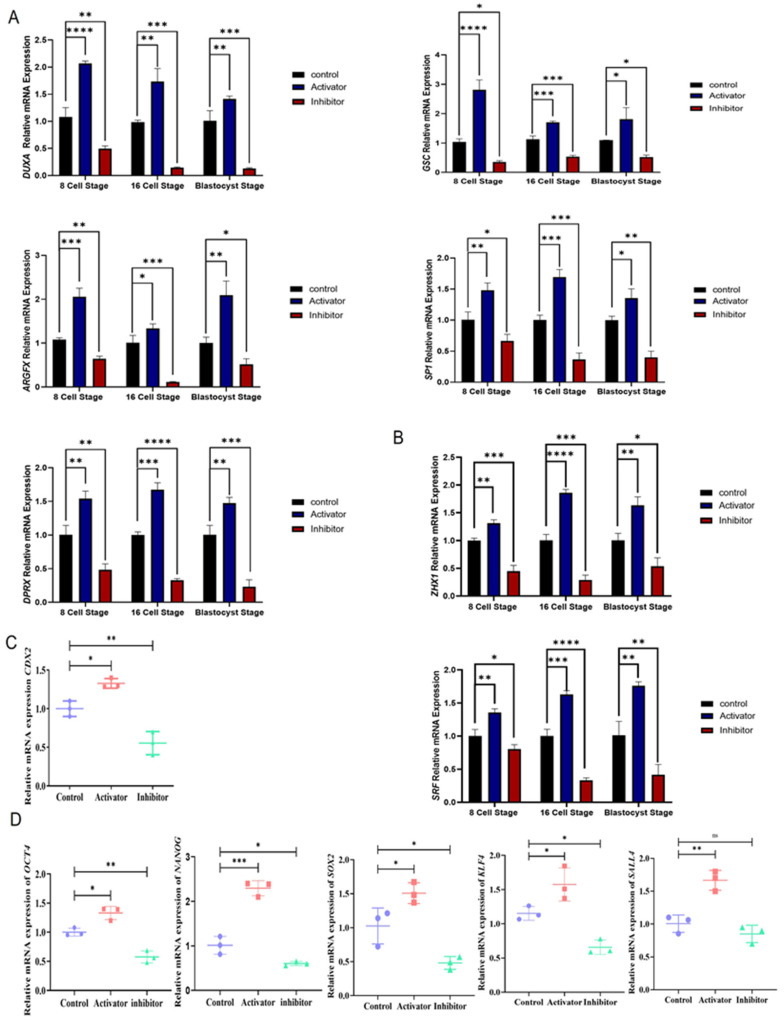
PDGFRβ and NFYA have been linked to zygotic genome activation. (**A**) Relative mRNA expression of zygotic genome activation related to *DUXA*, *GSC*, *ARGFX*, *SP1*, and *DPRX* genes in the control, activator, and inhibitor groups. (**B**) Relative mRNA expression of *SRF* and *ZXH1* in control, activator, and inhibitor groups. (**C**) Relative mRNA expression levels of trophoblast-related *CDX2.* (**D**) Relative mRNA expression levels of *OCT4*, *NANOG*, *SOX2*, *KLF4,* and *SALL4* pluripotency-related genes. Data are shown as the mean ± SEM. For mRNA analysis, 8-cell and 16-cell embryos, 15 per group, and 6 blastocysts per group were used in triplicates. ns = not significant. * *p <* 0.05, ** *p <* 0.01, *** *p <* 0.001, and **** *p <* 0.0001 denote significant differences. SEM—standard error of mean.

**Figure 4 ijms-24-17047-f004:**
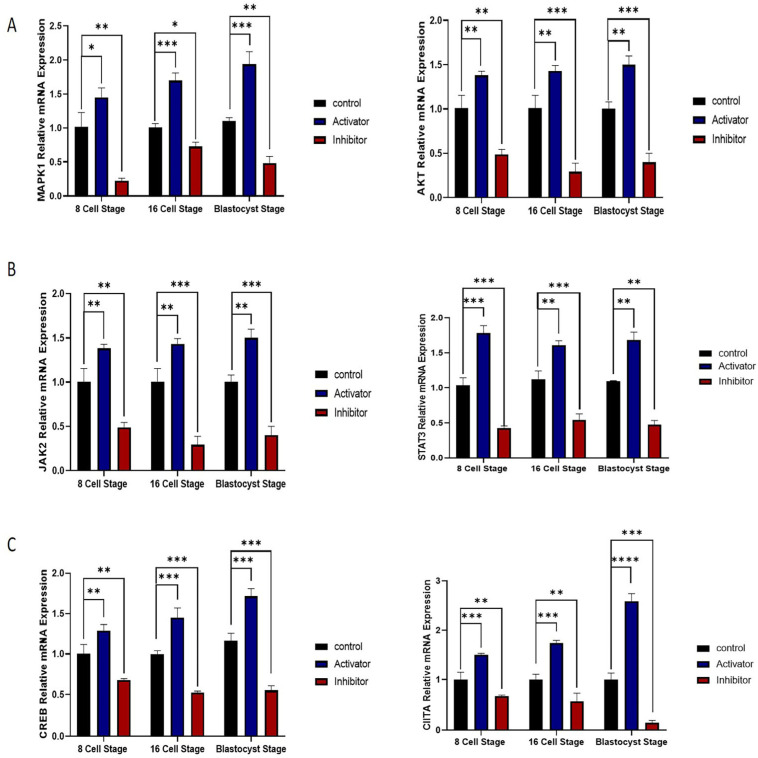
NFYA nuclear localization via PDGFRβ activation. (**A**). Relative mRNA expression of *MAPK1* and *AKT* in control, activator, and inhibitor groups. (**B**). Relative mRNA expression of *JAK2* and *STAT3.* (**C**) Relative mRNA expression of *CIITA*, *CREB* in control, activator, and inhibitor groups. For mRNA analysis, 8-cell and 16-cell embryos, 15 per group, and 6 blastocysts per group were used in triplicates. The experiments were repeated three times. Data are shown as the mean ± SEM. * *p <* 0.05, ** *p <* 0.01, *** *p <* 0.001, and **** *p <* 0.0001 denotes significant differences. SEM—standard error of mean.

**Figure 5 ijms-24-17047-f005:**
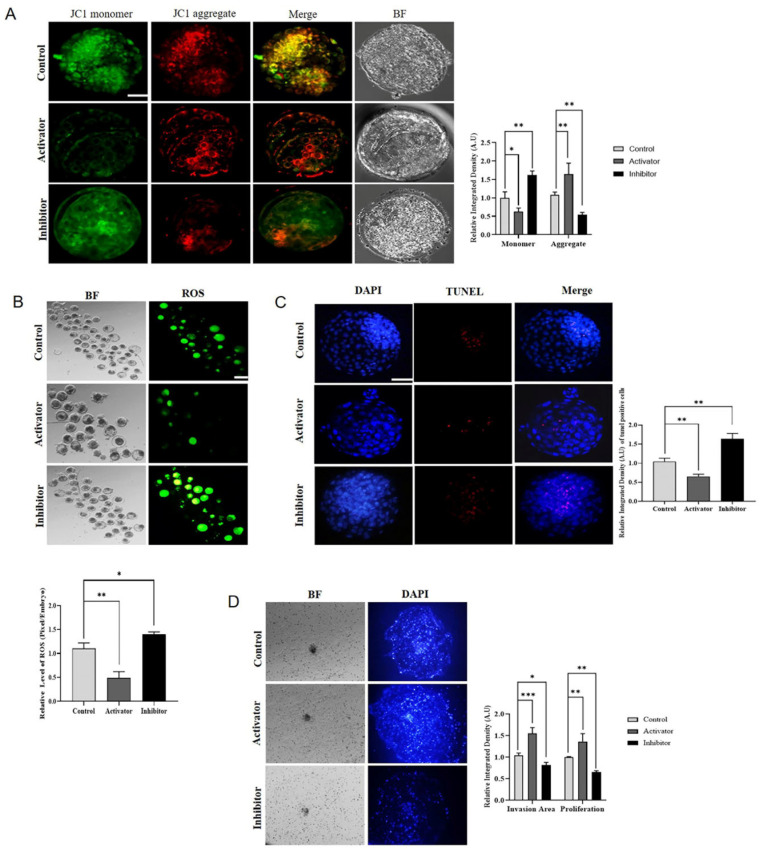
PDGFRβ and NFYA affect embryonic mitochondria and implantation potential of bovine embryos. (**A**) JC-1 staining of day-8 blastocyst control, activator, and inhibitor groups (n = 15 blastocysts per group). The scale bar indicates 20 µm. (**B**) H_2_DCFDA staining in embryos in control, activator, and inhibitor groups. The scale bar indicates 100 µm. (**C**) The TUNEL assay in day-8 blastocysts to examine apoptosis in control, activator, and inhibitor groups (n = 15 blastocysts per group). The scale bar indicates 20 µm. (**D**) The invasion area and cell proliferation of implanted blastocysts in the control, activator, and inhibitor groups (three blastocysts per group). The magnification is 40×. All the experiments were repeated three times. Data are shown as the mean ± SEM. * *p <* 0.05, ** *p <* 0.01, and *** *p <* 0.01 denote significant differences. SEM—standard error of mean. BF = Bright Field.

**Table 1 ijms-24-17047-t001:** The activation and inhibition effects of PDGFRβ on bovine embryonic development.

**Groups**	**No. of Presumed Zygotes**	**No. of Cleavage** **Embryo (% ± SEM)**	**No. of Blastocysts (% ± SEM)**	**No. of Hatched** **Blastocysts (% ± SEM)**
Control	240	190 (79.1 ± 1.32) ^a^	72 (30.0 ± 0.36) ^a^	21 (29.1 ± 1.26) ^a^
Activator	240	202 (84.2 ± 1.92) ^b^	93 (38.7 ± 1.25) ^b^	37 (39.7 ± 1.58) ^b^
Inhibitor	238	168 (70.5 ± 1.86) ^c^	40 (16.8 ± 0.57) ^c^	8 (20.0 ± 1.38) ^c^

^a,b,c^ Values with different superscripts in the same column are significantly different (*p* < 0.001). This experiment was completed in five replicates. SEM—standard error of mean.

**Table 2 ijms-24-17047-t002:** qRT-PCR Primer Information.

Name	Accession No.	Order Name	Sequence (5′-3′)	Product Size (bp)
*AKT*	NM_173986.2	F	CTCGCACGGAGGATCTGTAT	88
		R	TCCTCTCCATCCTGTGTTGG	
*ARGFX*	XM 024997266.1	F	GCTAGTGGCCTCAGTTCCTG	73
		R	GGAGGTGGTCACATAACGCA	
*CDX2*	NM_001206299.1	F	TGAGGAGCATGGACTCTGCTA	82
		R	GGGCTAGGTCAGCTGGTAAAC	
*DPRX*	XM 024979480.1	F	GCGTCCAGACTTGCACAAAG	169
		R	TGCCAACTGTTTCTCCGTGA	
*DUXA*	XM 024979492.1	F	GCCGTACCTCGTTCACAGAA	70
		R	CCAGGGTAAGGGTTTTGGCT	
*GSC*	NM_001192386.1	F	GACCAAGTACCCAGACGTGG	96
		R	TCTCAGCGTTTTCCGACTCC	
*KLF4*	NM_001105385.1	F	CCAACTACCCTCCCTTCCTG	73
		R	GGCATGAGCTCTTGGTAATGG	
*MAPK1*	NM_175793.2	F	AACAAAGTCCGAGTCGCCAT	194
		R	CGATGGTCGGTGCTCGAATA	
*NANOG*	NM_001025344.1	F	TGTGTCAATTTGAGGGAAGGGT	175
		R	ACTTTTGCCCCCTGTGCTTA	
*NFYA*	NM_001014956.1	F	GATTTGGAGGGGCCATGGAA	197
		R	CATTAATGGCTGCCCCTGGA	
*NFYB*	NM_001079786.2	F	TGTCCAACCAAACAGCCGAT	124
		R	AGAACCGTGTTGTCAGTGGT	
*NFYC*	NM_001034598.1	F	CAGAGGTCCAGCAAGGACAG	117
		R	ACTGGATGAACATGGGCTGG	
*OCT4*	NM_174580.3	F	GGGCAAACGATCAAGCAGTG	172
		R	CTCAGGGAATGGGACCGAAG	
*PDGFR* *α*	NM_001075896.2	F	CGCGGTTTTTGAGCCCATTA	95
		R	GGAAAAAGTGTGTCCACGGC	
*PDGFRβ*	NM_001075896.2	F	AAGGCATCAGCAGCAAGGAT	169
		R	GTGCTGGAGAGGTTGAGGAC	
*SP1*	NM_001078027.1	F	TGCTACCATGAGCGACCAAG	190
		R	CAAAGGGGATGGCTGGGATT	
*SRF*	NM_001206016.1	F	AGGCACTGATTCAGACCTGC	98
		R	TGTCTCTTCAAAGCCGGTGG	
*SOX2*	NM_001105463.2	F	GGCGGGGGAGACATTTTCAA	78
		R	AGCGTACCGGGTTTTTCTCC	
*SALL4*	NM_001192836.3	F	CGTCCAAGAAAGGCAAAGGG	127
		R	TGCAAGGAGCTATCAGTCCC	
*ZHX1*	NM_001205310.2	F	GTTTGAAATGCAAGATGGCGG	145
		R	TTTGACACGGAAGGGTGTCC	

F = Forward, R = Reverse.

## Data Availability

The data presented in this study are available on request.

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
