# Peer review of "PDGFRβ Activation Induced the Bovine Embryonic Genome Activation via Enhanced NFYA Nuclear Localization"

_ijms, 2023, doi:10.3390/ijms242317047_

Round 1

Reviewer 1 Report

Comments and Suggestions for Authors

A study by Perera et al. involves the effects of PDGFRβ modulation on embryonic genome activation and the possible involvement of the transcription factor NFYA in preimplantation bovine embryos. Though the paper is well-written, both the methodology and data presentation require significant improvements.

Major comments:

The rationale for this paper is not sufficient and require further consideration.

The authors claim that PDGFRβ activation results in nuclear localization of NFYA and the translocation of the latter factor affected the expression of several genes regulated by it. However, the nuclear localization of NFYA presented data in Figure 2A is partially convincing, authors are advised to revise this point. The other important point is the genes suggested to be activated upon NFYA activation is still doubtful unless the binding of the NFYA to the promoters of these genes using specific assays, e.g., CHIP. In other In other words, could the change in the expression of these genes be affected by PDGFRβ activation independently from NFYA?

The abstract partially reflects the paper contents.  For instance, line 22-24: Activation and inhibition of PDGFRβ using PDGF-BB and CP-673451 revealed that NFYA expression is significantly (p < 0.05) affected by the PDGFRβ. Please revise and be more specific. Also, line 24-26: Downregulation of NFYA following PDGFRβ inhibition significantly reduced (p < 0.05) the expression of critical EGA-related genes, bovine embryo development rate, and implantation potential. In view of lack of functional validations, the words “significantly reduced” could be changed to “was associated with”. The same applies to line 28.

The immunofluorescence data in Figure 1 should be transposed so that it match those in Figures 2 and 5.  

The discussion section should be deepened. Don’t cite figures within the discussion, you already did that in the result section.

The conclusions section (line 456-466) is nicely written. Authors may list future implications for this work.

Experimental design (4.1) could be summarized using simple a schematic timeline.

Minor comments:

Line 102: Figure 1. mRNA and Protein expression of PDGFRβ, PDGFRα and NFYA > Figure 1. Relative mRNA and protein expression NFYA, PDGFRβ, and PDGFRα. You have to match the order of the three factors in the legend with that of the three charts of Figure 1A.

Line 126-129: which concentrations caused these changes, please update the text.

Figure 2D: the parts of these figures could be reversed so that the activation results appear first.

Line 132: … in 8 cell, 16 cell and day-8 bovine blastocysts control, activator, and inhibitor groups > … in 8 cell embryos, 16 cell embryos, and day-8 bovine blastocysts of control, activator, and inhibitor groups

Line 138: immunofluorescent assay > immunofluorescent staining.

Footnote of Table 2 should be revised.

Line 143: 2.3. PDGFRβ and NFYA are Linked with Embryonic Genome Activation > Linked to

Labels of Figure 3C,D is too small for the human eye.

4.1. Experimental Design: Why did you decide to titrate these concentration ranges for both PDGF-BB and CP-673451. Update the text, if possible.

Line 367: Then the samples were incubated overnight with primary antibodies … write their names! BTW, 4.11. Antibodies should be merged with 4.4. Immunofluorescence Analysis.

Subtitles 4.9. mRNA Extraction of and Complementary DNA (cDNA) Synthesis and 4.10. Real‐Time Quantitative PCR (qRT‐PCR) Analysis should be merged too.

The font of Table 1 is too small. Also revise the numbering of tables within the manuscript.

Comments on the Quality of English Language

Moderate editing of the English language is suggested.

Author Response

Response to Reviewer 1

  1. Summary

A study by Perera et al. involves the effects of PDGFRβ modulation on embryonic genome activation and the possible involvement of the transcription factor NFYA in preimplantation bovine embryos. Though the paper is well-written, both the methodology and data presentation require significant improvements.

A) Thank you very much for taking the time to review this manuscript. Again, we thank the reviewers for their deep and thorough evaluation of our manuscript. Their scientific and constructive comments and remarks helped the value of this manuscript. Please find the detailed responses below and the corrections highlighted.

Point-by-point response to comments and suggestions for authors

Major comments:

Comment 1. The rationale for this paper is not sufficient and require further consideration.

A) To enhance the rationale of this paper, we improved the discussion section and Figure 1A section. Please find the changes in the revised manuscript – line number 270-276

Comment 2.  The authors claim that PDGFRβ activation results in nuclear localization of NFYA and the translocation of the latter factor affected the expression of several genes regulated by it. However, the nuclear localization of NFYA presented data in Figure 2A is partially convincing, authors are advised to revise this point.

A) We made some changes to Figure 2A to convince the nuclear localization of NFYA. Please find the changes in the revised manuscript. Page number 4.

The other important point is the genes suggested to be activated upon NFYA activation is still doubtful unless the binding of the NFYA to the promoters of these genes using specific assays, e.g., CHIP. In other in other words, could the change in the expression of these genes be affected by PDGFRβ activation independently from NFYA?

A) Previous studies have shown that SP1, GSC, DUXA, ARGFX, and DPRX involved in mammalian EGA (Bevilacqua et al., 2000; Bentsen et al., 2022; Laco et al., 2017; Otega et al., 2018). Furthermore, SP1 and DUXA directly interact with NFYA (Roder et al., 1999; Yamada et al., 2000; Eidahl et al., 2016). ARGFX and DPRX are newly identified transcription factors in EGA (Otega et al., 2018). Additionally, previous studies poorly understood the direct or indirect relationship between PDGFRβ activation and these genes independently from NFYA. Therefore, our study provides an approach to elucidate the upregulation of NFYA via PDGFRβ activation associated with the expression of these EGA genes.

References

Bevilacqua, A., Fiorenza, M.T. and Mangia, F., 2000. A developmentally regulated GAGA box-binding factor and Sp1 are required for transcription of the hsp70. 1 gene at the onset of mouse zygotic genome activation. Development127(7), pp.1541-1551.

Bentsen, M., Goymann, P., Schultheis, H., Klee, K., Petrova, A., Wiegandt, R., Fust, A., Preussner, J., Kuenne, C., Braun, T. and Kim, J., 2020. ATAC-seq footprinting unravels kinetics of transcription factor binding during zygotic genome activation. Nature communications11(1), p.4267.

De Iaco, A., Planet, E., Coluccio, A., Verp, S., Duc, J. and Trono, D., 2017. DUX-family transcription factors regulate zygotic genome activation in placental mammals. Nature genetics49(6), pp.941-945.

Ortega, N.M., Winblad, N., Reyes, A.P. and Lanner, F., 2018. Functional genetics of early human development. Current Opinion in Genetics & Development52, pp.1-6.

Roder, K., Wolf, S.S., Larkin, K.J. and Schweizer, M., 1999. Interaction between the two ubiquitously expressed transcription factors NF-Y and Sp1. Gene234(1), pp.61-69.

Yamada, K., Tanaka, T., Miyamoto, K. and Noguchi, T., 2000. Sp family members and nuclear factor-Y cooperatively stimulate transcription from the rat pyruvate kinase M gene distal promoter region via their direct interactions. Journal of Biological Chemistry275(24), pp.18129-18137.

Eidahl, J.O., Giesige, C.R., Domire, J.S., Wallace, L.M., Fowler, A.M., Guckes, S.M., Garwick-Coppens, S.E., Labhart, P. and Harper, S.Q., 2016. Mouse Dux is myotoxic and shares partial functional homology with its human paralog DUX4. Human molecular genetics25(20), pp.4577-4589.

Comment 3. The abstract partially reflects the paper contents.  For instance, line 22-24: Activation and inhibition of PDGFRβ using PDGF-BB and CP-673451 revealed that NFYA expression is significantly (p < 0.05) affected by the PDGFRβ. Please revise and be more specific. Also, line 24-26: Downregulation of NFYA following PDGFRβ inhibition significantly reduced (p < 0.05) the expression of critical EGA-related genes, bovine embryo development rate, and implantation potential. In view of lack of functional validations, the words “significantly reduced” could be changed to “was associated with”. The same applies to line 28.

A) Please find the revised manuscript’s changes in line 22-24 and 29.

Comment 4. The immunofluorescence data in Figure 1 should be transposed so that it matches those in Figures 2 and 5.  

A) Please find the changes in the revised manuscript – Figure 1. Page number 3

Comment 5. The discussion section should be deepened. Don’t cite figures within the discussion, you already did that in the result section.

A) The discussion section was improved (line numbers 270-276), and the figures within the discussion were removed.

Comment 6. The conclusions section (line 456-466) is nicely written. Authors may list future implications for this work.

A) Please find the changes in the revised manuscript – line number 504.

Comment 7. Experimental design (4.1) could be summarized using simple a schematic timeline.

A) Please find the changes in the revised manuscript – Experimental design (4.1).

Minor Comments:

Comment 1. Line 102: Figure 1. mRNA and Protein expression of PDGFRβ, PDGFRα and NFYA > Figure 1. Relative mRNA and protein expression NFYA, PDGFRβ, and PDGFRα. You have to match the order of the three factors in the legend with that of the three charts of Figure 1A.

A) Please find the revised manuscript's changes–line 114-119.

Comment 2. Line 126-129: which concentrations caused these changes, please update the text.

A) It has already mentioned in the experiment design 4.1.1 and 4.1.2. Please find it in line numbers 321 and 338.

Comment 3. Figure 2D: the parts of these figures could be reversed so that the activation results appear first.

A) Please find the changes in the revised manuscript – Figure 2D -line number 145-150.

Comment 4. Line 132: … in 8 cell, 16 cell and day-8 bovine blastocysts control, activator, and inhibitor groups > … in 8 cell embryos, 16 cell embryos, and day-8 bovine blastocysts of control, activator, and inhibitor groups

A) Please find the changes in the revised manuscript –line number 145.

Comment 5. Line 138: immunofluorescent assay > immunofluorescent staining.

A) Please find the changes in the revised manuscript –line number 152. 

Comment 6. Footnote of Table 2 should be revised.

A) Please find the changes in the revised manuscript –Table 1. Page number 5.

Comment 7. Line 143: 2.3. PDGFRβ and NFYA are Linked with Embryonic Genome Activation > Linked to Labels of Figure 3C, D is too small for the human eye.

A) Please find the changes in the revised manuscript. Line number 159 and page number 6 figure 3C, D.

Comment 8. 4.1. Experimental Design: Why did you decide to titrate these concentration ranges for both PDGF-BB and CP-673451. Update the text, if possible.

A) Thanks for pointing this out. We used these concentrations to determine the optimal range for our specific experiment. In addition, we did preliminary experiments using this range of concentration to identify the effects of different concentrations to avoid excessive or negligible responses.

Comment 9. Line 367: Then the samples were incubated overnight with primary antibodies … write their names! BTW, 4.11. Antibodies should be merged with 4.4. Immunofluorescence Analysis.

A) Please find the changes in the revised manuscript- line number 392,393. We deleted the 4.11 Antibodies section.

Comment 10. Subtitles 4.9. mRNA Extraction of and Complementary DNA (cDNA) Synthesis and 4.10. RealTime Quantitative PCR (qRTPCR) Analysis should be merged too.

A) Please find the changes in the revised manuscript. Line number 445.

Comment 11. The font of Table 1 is too small. Also revise the numbering of tables within the manuscript.

A) Please find the changes in the revised manuscript in page number 14.

Comment 12. Comments on the Quality of English Language Moderate editing of the English language is suggested.

A) The English editing company already checked this manuscript before submission.

Reviewer 2 Report

Comments and Suggestions for Authors

The paper “PDGFRβ Activation induced the Bovine Embryonic Genome Activation via enhanced NFYA Nuclear Localization” by Perera et al. describes the key role of NFYA in bovine embryonic genome activation and early embryonic development and its relationship with the platelet-derived growth factor receptor-β (PDGFRβ). Needless to say that there is a growing interest to understand the very complex and developmental process from zygote to the early embryo. There are many different interactions and cross talks between several factors, cells, structures and molecular modifications. Therefor the authors should define clearly the objectives of this study and adjust the title of the ms. Both, the Introduction and Results have been shortened, and the Discussion has also been edited to remove some of the less important material. Nevertheless it is a very well designed study focussing on an important mechanism of the EGA with some clear and impressive results. However, some results are overstated (implantation) and do not support strong conclusions as reported in the discussion. In general, this is a very informative and scientific significant manuscript. Overall, it is well written, but I do have a few suggestions for improvement:

Major concerns:

1)    How do the authors classify their blastocyst rate of 30 % in the control group in comparison to international standards?

2)    How do the authors interpret their findings regarding the implantation potential without evaluating the hatching rate and a real implantation rate?

3)    How would the authors characterize the invasion area without any reflection to control measurements and control groups (in vivo experiments)? Especially the observation of hatched blastocyst in vitro needs a comparison with in vivo embryos.

Minor concerns:

1)    There is no detailed and coherent description of the complete IVC processing ( L531). See Nr. 1 above. Invasion area was measured on day ten ? How are the hatching rates in these experimental groups.

2)    Figures 2 and 3 are hard to read and recognize.

3)    Table 1 hard to read. Positioning of Table 1 and 2 is confusing.

Author Response

Response to Reviewer 2 

  1. Summary

The paper “PDGFRβ Activation induced the Bovine Embryonic Genome Activation via enhanced NFYA Nuclear Localization” by Perera et al. describes the key role of NFYA in bovine embryonic genome activation and early embryonic development and its relationship with the platelet-derived growth factor receptor-β (PDGFRβ). Needless to say, that there is a growing interest to understand the very complex and developmental process from zygote to the early embryo. There are many different interactions and cross talks between several factors, cells, structures and molecular modifications. Therefor the authors should define clearly the objectives of this study and adjust the title of the ms. Both, the Introduction and Results have been shortened, and the Discussion has also been edited to remove some of the less important material. Nevertheless, it is a very well-designed study focussing on an important mechanism of the EGA with some clear and impressive results. However, some results are overstated (implantation) and do not support strong conclusions as reported in the discussion. In general, this is a very informative and scientific significant manuscript. Overall, it is well written, but I do have a few suggestions for improvement:

A) Thank you very much for taking the time to review this manuscript. Again, we thank the reviewers for their deep and thorough evaluation of our manuscript. Their scientific and constructive comments and remarks helped enhance the value of this manuscript. Please find the detailed responses below and the corrections highlighted. The introduction and discussion sections improved - line number 48-54 and 270-276. Objectives are mentioned in the abstract and introduction section. Line numbers 17-20, 87.

Major concerns:

Comment 1. How do the authors classify their blastocyst rate of 30 % in the control group in comparison to international standards?

A) According to the previous study, the average rate of oocyte development in vitro to the blastocyst stage is 30% (Ferronato et al., 2023).  In vitro blastocyst developmental rate is typically only about 20-40% (Tribulo et al., 2006). However, specific developmental rate standards can be varied according to several factors, such as laboratory conditions, culture media, embryo quality, and time in culture. According to our laboratory conditions, the blastocyst developmental rate averages 28-33% (Idrees et al., 2019; Wei et al., 2022).

References

Ferronato, Giuliana de Avila, Carolina Mônica Dos Santos, Paola Maria da S. Rosa, Alessandra Bridi, Felipe Perecin, Flávio Vieira Meirelles, Juliano Rodrigues Sangalli, and Juliano Coelho da Silveira. "Bovine in vitro oocyte maturation and embryo culture in liquid marbles 3D culture system." Plos one 18, no. 4 (2023).

Tríbulo, P., Rivera, R.M., Ortega Obando, M.S., Jannaman, E.A. and Hansen, P.J., 2019. Production and culture of the bovine embryo. Comparative embryo culture: methods and protocols, pp.115-129.

Idrees, M., Xu, L., Song, S.H., Joo, M.D., Lee, K.L., Muhammad, T., El Sheikh, M., Sidrat, T. and Kong, I.K., 2019. PTPN11 (SHP2) is indispensable for growth factors and cytokine signal transduction during bovine oocyte maturation and blastocyst development. Cells8(10), p.1272.

Wei, Y., Idrees, M., Sidrat, T., Joo, M., Xu, L., Ko, J. and Kong, I., 2022. BOEC–Exo Addition Promotes In Vitro Maturation of Bovine Oocyte and Enhances the Developmental Competence of Early Embryos. Animals 2022, 12, 424.

Comment 2. How do the authors interpret their findings regarding the implantation potential without evaluating the hatching rate and a real implantation rate?

A) We evaluated the hatching rate during the study. The hatching rate can be found in Table 1 in the revised manuscript. We compared our hatching rate with previous in vitro studies, and it aligned with previous study results (Park et al., 1999). Evaluation of real implantation potential is an important measure. However, we didn’t perform the in vivo analysis in this study. Therefore, we performed an invasion chamber assay to evaluate blastocyst implantation potential through a simulated extracellular matrix (Matrigel) (Wei et al., 2022). These assays cannot be compared directly to in vivo embryos. However, it can provide insights into the blastocyst implantation potential under specific experimental conditions. In addition, we used high-quality blastocysts to consider blastocoel expansion and inner cell mass morphology. Because generally considered high-quality blastocysts have high implantation potential. Moreover, inner cell mass (ICM) grade is considered more important for determining the implantation potential of a blastocyst (Masuda et al., 2021). In this study, we evaluated expression levels of ICM-related genes (OCT4, NANOG, SOX2, KLF4, SALL4). Based on these results, we interpret our findings associated with the implantation of bovine embryos.

References

Park, S., Kim, E.Y., Yoon, S.H., Chung, K.S. and Lim, J.H., 1999. Enhanced hatching rate of bovine IVM/IVF/IVC blastocysts using a 1.48-μm diode laser beam. Journal of assisted reproduction and genetics16, pp.97-101.

Wei, Y., Idrees, M., Sidrat, T., Joo, M., Xu, L., Ko, J. and Kong, I., 2022. BOEC–Exo Addition Promotes In Vitro Maturation of Bovine Oocyte and Enhances the Developmental Competence of Early Embryos. Animals 2022, 12, 424.

Masuda, Y., Hasebe, R., Kuromi, Y., Kobayashi, M., Urataki, K., Hishinuma, M., Ohbayashi, T. and Nishimura, R., 2021. Three-dimensional live imaging of bovine preimplantation embryos: a new method for IVF embryo evaluation. Frontiers in Veterinary Science8,

Comment 3. How would the authors characterize the invasion area without any reflection to control measurements and control groups (in vivo experiments)? Especially the observation of hatched blastocyst in vitro needs a comparison with in vivo embryos.

A) It is commonly accepted that the developmental competence and quality of in vitro-produced embryos are overall lower than those of in vivo-produced embryos (Rabel et al., 2023). Iwasaki et al., 1990, showed that in vitro-produced embryos have a lower cell number in the inner cell mass of hatched blastocysts than in vivo-produced embryos. However, previous studies that evaluated the invasion area using in vitro studies align with our study (Khan et al., 2023; Idrees et al., 2019).

Khan, A.M., Idrees, M., Perera, C.D., Haider, Z., Joo, M.D., Kang, J.S., Lee, S.H. and Kong, I.K., 2023. The effects of cycloastragenol on bovine embryo development, implantation potential and telomerase activity. Reproduction, Fertility and Development.

Idrees, M., Xu, L., El Sheikh, M., Sidrat, T., Song, S.H., Joo, M.D., Lee, K.L. and Kong, I.K., 2019. The PPARδ agonist GW501516 improves lipolytic/lipogenic balance through CPT1 and PEPCK during the development of pre-implantation bovine embryos. International Journal of Molecular Sciences20(23), p.6066.

Minor concerns:

Comment 1. There is no detailed and coherent description of the complete IVC processing (L531). See Nr. 1 above. Invasion area was measured on day ten? How are the hatching rates in these experimental groups?

A) We changed the L521 because L521 shows the reference for the IVC procedure - line number 645. The invasion area was measured after ten days of invasion assay, not on day ten. Hatching rates in these experimental groups can be found in the revised manuscript Table 1.

Comment 2. Figures 2 and 3 are hard to read and recognize.

A) Please find the changes in the revised manuscript. Pages number 4 and 6

Comment 3. Table 1 hard to read. Positioning of Table 1 and 2 is confusing.

A) Please find the changes in the revised manuscript. Page number14. Line number 139 and 470.

Round 2

Reviewer 1 Report

Comments and Suggestions for Authors

The authors have addressed my concerns. Pending editorial approval, this version of the manuscript could be accepted.